# Embedding Weather Simulation in Auto-Labelling Pipelines Improves Vehicle Detection in Adverse Conditions

**DOI:** 10.3390/s22228855

**Published:** 2022-11-16

**Authors:** George Broughton, Jiří Janota, Jan Blaha, Tomáš Rouček, Maxim Simon, Tomáš Vintr, Tao Yang, Zhi Yan, Tomáš Krajník

**Affiliations:** 1Artificial Intelligence Center, Faculty of Electrical Engineering, Czech Technical University in Prague, 166 27 Prague 6, Czech Republic; 2Unmanned System Research Institute, Northwestern Polytechnical University, Xi’an 710072, China; 3CIAD, UBFC, Université de Technologie de Belfort-Montbéliard, F-90010 Belfort, France

**Keywords:** long-term autonomy, machine learning, self-supervised learning, inclement weather conditions

## Abstract

The performance of deep learning-based detection methods has made them an attractive option for robotic perception. However, their training typically requires large volumes of data containing all the various situations the robots may potentially encounter during their routine operation. Thus, the workforce required for data collection and annotation is a significant bottleneck when deploying robots in the real world. This applies especially to outdoor deployments, where robots have to face various adverse weather conditions. We present a method that allows an independent car tansporter to train its neural networks for vehicle detection without human supervision or annotation. We provide the robot with a hand-coded algorithm for detecting cars in LiDAR scans in favourable weather conditions and complement this algorithm with a tracking method and a weather simulator. As the robot traverses its environment, it can collect data samples, which can be subsequently processed into training samples for the neural networks. As the tracking method is applied offline, it can exploit the detections made both before the currently processed scan and any subsequent future detections of the current scene, meaning the quality of annotations is in excess of those of the raw detections. Along with the acquisition of the labels, the weather simulator is able to alter the raw sensory data, which are then fed into the neural network together with the labels. We show how this pipeline, being run in an offline fashion, can exploit off-the-shelf weather simulation for the auto-labelling training scheme in a simulator-in-the-loop manner. We show how such a framework produces an effective detector and how the weather simulator-in-the-loop is beneficial for the robustness of the detector. Thus, our automatic data annotation pipeline significantly reduces not only the data annotation but also the data collection effort. This allows the integration of deep learning algorithms into existing robotic systems without the need for tedious data annotation and collection in all possible situations. Moreover, the method provides annotated datasets that can be used to develop other methods. To promote the reproducibility of our research, we provide our datasets, codes and models online.

## 1. Introduction

The reliability of robotic perception is directly derived from the quality of the data used to train, tune, and develop its modules, often in the form of neural networks. There are two significant issues when creating datasets appropriate for outdoor robotics, which can prevent these algorithms from attaining their desired characteristics.The first is the issue of humans labelling the data, and the second is that often, the datasets used are not representative enough of less common weather conditions.

As most of the current machine learning models used in robotic perception fall under the category of supervised learning, they need annotation labels for the dataset used to train them. The process of creating such datasets is quite challenging and requires a huge amount of human effort. For perspective, a commonly used image dataset, Imagenet, consists of 3.2 million images with annotations [1]. To solve the problem of human error and ambiguity during the annotation process, they had each of these images human-annotated many times and looked for a consensus.

Creating a dataset for an outdoor robot is made especially complicated by the necessity of making it representative of things in a variety of conditions, such as weather, that a robot will encounter during its lifetime of operation. Some weather conditions such as rain, snowfall, or fog are less frequent or outright rare, which often reduces the number of relevant training samples collected and consequently the ability of the trained perception system to perform under such conditions. The under-representation of severe weather in datasets is further amplified by the natural behaviour of the data collectors, who tend to occupy areas with favourable climates and seek shelter rather than gather data during worsening weather—which applies to night conditions as well. As an example, consider the popular 2017 COCO validation image dataset [2] used to evaluate common data-hungry neural networks—only about 2% of the images were taken under rainy conditions, 0.5% were taken in snow, and only a single picture was taken in (light) fog.

In addition, severe weather conditions cause technical problems that make data collection more complicated. Low temperatures can cause batteries not to work properly, humidity can cause short circuits and corrosion, and the overall risk of system failure during extreme weather increases. Moreover, severe weather conditions can reduce the quality of the raw data. This can make the annotation process also suffer and, subsequently, significantly impair the performance of trained detectors in those situations. As an example, for LiDAR, the reflectivity of laser measurements is negatively affected if there is fog or snow present in the laser’s path, which can, therefore, distort the incoming data and impair algorithms working on this data.

Mobile systems, such as autonomous cars, are becoming increasingly widespread. They are inherently dependent upon their sensors and abilities of perception to operate autonomously in the world around them. As rare or adverse weather conditions can already be adversarial to hardware, poorly performing detectors can exacerbate the situation up to the point of property damage and health hazards. We can reason that less common weather events, therefore, are where our system is most vulnerable and are where our attention should be focused. Deficiencies in the perceptual abilities of autonomous vehicles have already been linked to several incidents in recent years [3,4], which underscore this topic as of critical importance.

One proposed solution to neural networks’ weather robustness is to have developers recreate massive datasets with a more balanced distribution of environmental conditions, as suggested by [5]. This is problematic twofold, as some conditions are simply unrealistic to achieve, as they are extremely unlikely, or the required manpower involved with the acquisition and painstaking human labelling of large amounts of data is simply too high. Certain weather conditions might require waiting for different parts of the year before work can commence; others might appear randomly in some area of size which cannot be easily covered, so the equipment would have to be moved extremely fast, which is difficult and further makes the chance of error higher. All of this adds to the time and costs associated with collecting the data. While some of these undertakings may be possible for very large projects or organisations, it becomes infeasible for smaller groups and individuals, harming the accessibility of robotics to newcomers, the development of the scientific field, and the evolution of innovative, rather than corporate, industry.

A possible way to avoid the rarity of such weather conditions is to use physical weather simulators. These place the sensors in an enclosed space where the weather can actually be controlled. Although the cost of these is high, they enable the collection of otherwise unattainable data. Unfortunately, most research groups cannot afford them; they still present a barrier between real and simulated conditions and, compared to the method proposed in this article, cannot be performed onboard during the actual operation for which the robot was designed.

Existing pieces of literature have looked at techniques from which the automatic labelling of data can be achieved. While helping to address the costs associated with the labelling of data, both financial and time-wise, this would still mean having data collectors working over an extended period of time to actually gather the data from varied conditions. Furthermore, when poor-weather data is collected, in our experience, auto-labelling techniques work less effectively on this data.

In this article, we approach the problem of training detectors in heterogeneous weather conditions holistically. We show how an auto-labelling-based scheme can be deployed on a robot to train a neural network for car detection, where we employ a simulator-in-the-loop approach for ensuring good weather robustness. This way, we can generate trained detector models robust to inclement weather conditions, while importantly, for training data, having only ever supplied raw robotic sensor data from regular environmental conditions.

Our pipeline makes use of a clustering-based hand-coded detection method to supply initial candidate detections of cars from a moving car transporter platform. These candidates are then refined and corrected through the use of temporal filters to provide annotation labels for the LiDAR scene. With the data labels extracted, the scene is run through the weather simulations, and these altered scenes with the labels are passed to the network for training. We show how this approach is able to generate trained neural networks that can operate effectively, despite no human-labelled or weather-varied training data.

In this paper, we propose a proof-of-concept on how to tackle the problems outlined above from the point of view of robotic research groups. We needed to build a system capable of detecting surrounding cars in planar LiDAR scans to operate autonomously in an outdoor car park in all seasons. There was no publicly available datataset with sufficient size and weather condition diversity that would match our specific sensory setup. The effort of collecting a sufficiently diverse dataset and labelling it using conventional methods would consume most of the resources allocated for the car transporter project. The design and deployment of the simulator-in-the-loop self-annotating pipeline already saved significant manpower associated with data collection and annotation.

## 2. Related Work

Work on detecting cars in planar LiDAR data has previously looked for the characteristic “L” shape visible from the front and side of cars in laser scans [6,7,8,9]. An approach based on clustering, segmentation, and Bayesian programming has been applied for this detection task [10], although such approaches can be vulnerable to detecting the corners of buildings, for example, as vehicles. Others have applied SVM-based methods to the problem of finding such objects in scans [11,12,13,14] or by using Adaboost [15,16].

As a relevant alternative to LiDAR, radar sensors have also seen much research for the purpose of detecting cars due to the fact that the sensors can be neatly integrated into a car’s body; that is to say, they can “see through” plastic bumpers and thus be hidden behind them. Similar to LiDAR, custom features have been used to find cars and people in radar signals [17,18,19]. The works include the application of random forest classifiers to occupancy grids [20] for car detection. Lombacher et al. extended this by accumulating radar frames and then hand labelling them and using them to train a neural network to increase the robustness of car detection [21] and later extended this to other classes within a static environment [22]. Danzer et al. combined the acts of classification and bounding boxes into a single detector [23].

However, all these techniques for finding cars that involve deep learning neural networks, while increasing robustness, require human annotation. The existing work has looked at how we can take advantage of multiple data streams to self-train neural networks. Blum and Mitchell developed a method by which they had a pair of *views* of their data [24]. They trained a classifier for the textual component of a web page and another classifier on the metadata around hyperlinks to that web page. Given that either *view* of the data in isolation is *probably* sufficient to classify the web page, they can be independently trained on a small subset of the data and can then predict labels on new unlabelled data to increase the training set of the other. Qiao et al. built upon this and developed a co-learning framework for deep neural networks that took a pair of images from different image domains (infrared, ultraviolet, etc.) [25]. The image pairs are used to train multiple neural networks with the constraint that the neural networks should not become too similar. In a similar vein, other work in the field of auto-labelling data has made use of a multi-model approach; for example, in 2008, Moxley et al. applied an unsupervised based method to apply labels to videos by exploiting both the visual and transcript of a video where the transcript was automatically produced, and the combination of these was used to mine keywords [26].

Of course, such techniques generally involve noisy training data, which might negatively impact the training process. Unless a human has checked every label by hand (and perhaps even if so), it is unlikely that the labels will be perfect. Fortunately, the stochastic learning process of neural networks is inherently resilient to a certain amount of noise. However, a common technique to combat this is the use of a trusted subset of data known to be well annotated due to human effort [27] or additional automated annotation quality checks [28]. This can be used to perform some initial training before switching to training on the full dataset. Other work has shown that problems with training networks from noisy data can be reduced if the stochastic noise model is known a priori [29]. Often these two problems are solved together by "polishing" a subset and then analysing how the labels compare to estimate the noise. While this may alleviate some of the problems of data annotation, unfortunately, it does not solve it, but these works show it is not necessary to have a dataset completely free of noise.

In 2021, Qi et al. showed how a pipeline could be constructed by taking an already trained neural network and using it to find similar objects in real-world data [30]. By doing so, they can generate additional labels for real-world point clouds with little human effort and, in turn, use these to train further neural networks, although they do not address the training of the initial network.

Nevertheless, previous research has shown the automatic training of neural networks, especially by utilising robots. The authors of [31] examined the feasibility of using a simple method to automatically provide training samples to a neural network and showed that the neural network could outperform the method from which it was trained. Similarly, Chadwick and Newman looked at using radar to automatically annotate images for a deep learning neural network [32]. However, they did not address how to identify targets in the incoming data. Instead, they assumed that all moving radar targets over a given threshold speed (after subtracting ego-motion) were a vehicle. Additionally, they identified one of the benefits of using radar as its ability to make velocity estimates of the target from a single data point. This means that they do not have to track target points between several frames to identify if they are moving. While a nice feature for data processing, it has the consequence that they were unable to detect tangential movement in the radar. As a result, these targets were not identified as cars and possibly incorrectly treated as negative training samples. This resulted in the need to apply techniques to processing noisy training data [33].

Without tracking, they lost the ability to exploit the temporal information about the target, information that was so valuable to [30]. Qi et al. used a custom implementation of a multi-object tracker from [34] in order to experiment with accumulating LiDAR frames and to feed multiple frames into a network for context at the same time. Similarly, the temporal filter was crucial for Chen et al. [35] using a similar conceptual pipeline as [30]; they used it to identify and label moving objects within point clouds, from which they trained and evaluated a neural network.

In the field of auto-labelling, a common approach is to use a trivial teaching technique to extract labels and then use this to train a teachable method, commonly in the form of a neural network. Hansen et al. extracted labels of humans in sets of images captured by a surveillance camera. In this example, the teaching technique was using background subtraction, along with a body model, to extract these labels. Similarly, in [36], the authors paired a radar sensor with a LiDAR sensor and used well-known LiDAR people detection methods to annotate where people were present in the radar data. Using these as training samples, they taught the neural network to recognise people in radar data.

Although many of these works have looked at how to detect objects, prior work has shown the difficulty of simply navigating a robot around known environments. Autonomous navigation in the outdoors can become significantly more difficult as environmental conditions deteriorate [37,38]. Li et al. examined foggy weather conditions and their impact on LiDAR sensors from an automotive perspective [39]. This was then used to train a neural network to predict the minimum distance at which successful detections could be made in the current fog conditions. Others have tried to benchmark the performance of these sensors under various conditions [40,41].

To counteract this, work has been done to make the detectors themselves more robust to weather conditions, despite shortcomings in the original training datasets. In 2020, Yang et al. published a method to simulate foggy conditions within the collected LiDAR data [42]. Moreover, Hahner et al. showed various simulators for fog and snowy conditions and how these could be used on large datasets as a form data augmentation to extend the datasets [43,44].

## 3. Problem Definition

Previous work has hinted at the promise of auto-labelling-based techniques to realise trained neural networks without the effort of human labelling. This is based on the key ability of neural networks to learn to solve non-trivial tasks without specifically engineered algorithms and hand-tuned parameters. Therefore, as long as we can provide good training samples to the networks, humans do not need to participate directly in the development of the training dataset.

We apply this idea to the task of developing a car detector for an autonomous robot. As the robot traverses its environment, it can collect data and automatically propose training samples to the network. By doing so, as in previous works, we can develop a well-trained neural network detector without human labelling. This is advantageous because there is a potentially infinite pool of training samples available, therefore covering a wider range of situations and conditions than hand-developed training datasets.

The downside, however, of having each detector develop its own dataset is that unless the robot is actually deployed for long periods of time, it may not encounter less common weather conditions from which to develop training samples. We propose as a solution that for such auto-labelling pipelines, the inclusion of publicly available weather simulation models should be a part of the training loop. As our robot collects data, the training set developed for the neural network is immediately balanced with regards to outdoor weather conditions—whether or not the robot has actually seen those conditions—thus helping with the robustness of the trained detector.

### Conceptual Overview

The aim of this article is to show how the key ideas from the previous section can be combined to train a detector that can work in heterogeneous conditions in an outdoor environment with limited resources of time, money and human effort. We propose a simulator-in-the-loop approach, whereby the weather simulator forms an integral part of the auto-labelling process.

We receive a set of LiDAR points, $P_{t}$, for each moment in time that the robot is operational, and we want to find the corresponding set of car detections, $D_{t}$, for that LiDAR frame. To form our auto-labelling pipeline, we start from within an initial source of information to provide candidate detections to our system. Where previous research has used mainly velocity to isolate dynamic agents in the surroundings [32,35], we opted for an approach based on our previous work [31], where a clustering approach based off of a state-of-the-art two-dimensional car detector was used to find candidate vehicles in the LiDAR scans. The approach consisted of using clusters in the planar scan taken close to the ground, representing the wheels of cars. The main idea of this was to provide some source of input objects against which to train, where the set of labels can then be processed by an offline temporal filter; however, this input method could easily be substituted for similar schemes, such as those used above.

With some provisional labels associated with the laser scan frame, the offline temporal filters can be utilised to form an improved set of labels by exploiting the temporal domain, $D_{t}^{{}^{\prime }}$. As the training process runs offline, the filter can search the dataset for any future or past detections. The offline temporal filter is also aware of the robot’s ego-motion, and therefore, if the robot moves out of sight of a car and then returns, there is still an ability to associate the new detections with the previous detections. As such, the temporal filter can build up a full picture of how that particular object has been detected at various points in time across the whole dataset and decide to classify it as a car or not. If the object is marked as a car, then the temporal filter can traverse the dataset frame by frame and record the object’s position in that frame, even if the original detector was not able to make a detection. Thereby, we can construct a dataset not limited to the original detector but in excess of the quality of the raw detector output. Similarly, any rogue detections that do not satisfy the temporal filter can be removed from the dataset, removing false detections.

While previous work has shown that a neural network can receive $P_{t}$ with $D_{t}^{{}^{\prime }}$ as annotations and learn to detect cars from this improved label set, we show that $P_{t}$ can undergo an additional step of being passed through weather simulations in order to increase the robustness of our network. With the set of refined set of labels already collected, we can then take each laser scan associated with that set of labels and modify it without affecting the labels. This means that we can involve the weather simulators to alter the scene. For example, artificial fog can be added to the laser scan. The corrected set of labels and the modified laser scans can then be recombined, and the data can be annotated according to each specific neural network input requirement. By such a process, we aim to develop neural networks that are trained and robust to poor weather, while having never received a human-annotated training sample and without being focused on bad weather data collection.

## 4. System Description

In the following, we look at the steps taken to realise a trained network for later evaluation, covering the robotic hardware and datasets captured, as well as the operation of each part of the pipeline.

### 4.1. Hardware

To evaluate our proposed pipeline, we collected data from a car transporter robot operating in a large car park. The robot, visible in Figure 1, is a large fixed-geometry Ackermann steering vehicle capable of autonomous operation for extended periods. It is equipped with three ground-level SICK MICS3 laser scanners, one in the front and one on either side. The robot is also equipped with one slightly higher SICK LMS111 laser scanner at the rear. The MICS3 laser scanners have a maximum range of 40 m, a frequency of 10 Hz, and a field of view of 275°, of which only 270° was used for the sides and 180° for the front. The LMS111 has a maximum range of 20 m, a frequency of 25 Hz, and a field of view of 270°, of which only 200° was used. The fields of view for each sensor were aligned to the cardinal directions.

### 4.2. Data Acquisition

During the course of a year, the robot was regularly deployed, and data were collected from its LiDAR sensors in a large car park. The parking covered an area of approximately four hectares in regular use. Data were collected during all weather conditions, including fog, hail, and heavy snow. The data covered the extremes of the local humid continental climate according to the Köppen taxonomy. Furthermore, data were collected at all times of the day, including day, night, dusk, and dawn. The other vehicles in the parking lot were generally static for safety reasons during the robotic operation, although there was some light traffic during data collection.

Each LiDAR sensor of the car transporter provides us with point clouds, where we index the sensors by S={1,2,3,4}. We denote the point clouds provided by these sensors as $P_{s,t}={\left\{{\left(x,y,i\right)}_{i}\right\}}_{i=1}^{|P_{s,t}|}$, where s∈S, *t* denotes the timestamp, and *i* denotes the LiDAR beam intensity. The point clouds are then temporally synchronised using the built-in methods for the robotic operating system (ROS). Each point cloud $P_{s,t}$ is then projected into a common robot-centric co-ordinate system, given by:$$P_{s,t}^{{}^{\prime }}=\left\{T_{s}\cdot {\left[x_{i},y_{i},1\right]}^{T},\forall {\left(x,y\right)}_{i}\in P_{s,t}\right\}$$
where $T_{s}$ is the transformation matrix from the sensor’s position relative to the vehicle. For conciseness, from here onwards, we will refer to $P_{s,t}^{{}^{\prime }}$ as $P_{s,t}$. We assume that cars present in $P_{s,t}$ are not *significantly* distorted by progressive scan effects from ego-motion for our experiment. Nevertheless, to verify this assumption, we performed and evaluated a brief experiment on moving vehicles described in Section 5.1.2.

For the purposes of evaluating the system performance, a subset of temporally contiguous data was segregated from the main set, with the ground truth for each data frame acquired by hand annotation. The annotation process consisted of negating the motion of the robot so that all scans accumulate and align into the map frame for each drive. A human was then tasked with marking where the cars are in the accumulated map, visible in Figure 2, and then marking which of the cars are visible in a frame-by-frame manner. The car’s *x* and *y* coordinates, as well as its yaw, ϕ, in the LiDAR frame of reference were marked for each scan. Due to the difficulty in telling the front from the back of a car in a planar laser scan, the yaw of the cars is ambiguous by 180°. The authors feel that this is not a significant issue for a detector, as when the cars are static, knowing the location and ambiguous orientation is sufficient for most purposes, and for moving cars, tracking information can be used to obtain the likely orientation. The process of hand-annotating the dataset took approximately 7 min on average per one minute of real data, illustrating the necessity for automatic machine labelling tools, as to annotate the entire dataset collected would have taken infeasible amounts of time.

### 4.3. Auto-Labelling Pipeline

For our pipeline, we allowed the robot to traverse its environment and collect data and conducted the data processing and network training afterwards. A visualisation of the important stages of the training pipeline can be seen in Figure 3.

#### 4.3.1. Clustering-Based Car Detector

To find the initial candidate detections of cars in the LiDAR frame, we used a clustering-based approach. As mentioned, the LiDAR sensor data are synchronised, and the data from each LiDAR are transformed into a common coordinate system. The points in the laser scan are passed through a density-based clustering algorithm—in the case of a successful detection of a car, these clusters generally correspond to the car wheels; see Figure 4. We then sample the set of wheel clusters, looking for situations where there are four wheels corresponding to the same geometry as the car we intend to detect. This has the added benefit that the parameters of the wheel spacing can be easily adjusted should we wish to add alternative car geometries to the dataset.

As we are aiming for a pipeline which saves on the development and annotation time necessary for an object detector, it would not make sense for this initial detection method to be complex and time-consuming to implement, as that just moves the volume of work required from annotation to another part of the system. The clustering-based method we use here is lightweight, easy to implement, and substitutable for other methods for this reason.

The primary concern with the clusterer as a detector lies in its robustness. Clustering LiDAR points creates many clusters all over the scene, while poor weather and geometric occlusion of the scene can mean many legitimate wheel clusters remain undetected. Furthermore, at long range, the limited angular resolution of laser scanners results in distant vehicles having very few detected points, resulting in clustering rejections, which significantly increases the number of false positives. Even a range-aware clustering method would struggle, as poor weather conditions can create false LiDAR points, which become numerically significant. Finally, objects positioned next to wheels can become included in the cluster, which will drag the centre of the cluster away from the wheel and potentially cause the set of four wheels to no longer fit the required geometric model to be recognised as a car.

The method makes use of the DBScan algorithm [45] for its clustering of LiDAR points into wheels. We used an ϵ-neighbourhood of 0.35 m, appropriate for the size of a wheel, and a minimum cluster size of 5 points, which provided a good compromise between false positives and false negatives. It should be noted that a minimum of five LiDAR points per cluster necessarily provides a hard limit on the maximum range of the clustering-based method.

The wheel clusters were then analysed geometrically to find cars. As the training process is run offline, we can exhaustively sample for pairs of clusters and look for a pair 1.5±0.3 m apart, corresponding to the width of the vehicles. When a pair was found within this tolerance, we then looked for additional wheels by treating the first two positions as a line (corresponding to an axle of the car). We can then obtain the normal of the line at each wheel and then search for further clusters along the normals at the positions where we would expect the other two wheels to be. This gives us two possible scenarios where the car lies on one side or the other from the initial pair. If a further two wheels are found on the same side of the initial pair, 2.5±0.3 m along each of the normals, then we have found four wheels approximately correctly positioned as a car. As a final test, if the candidate car has made it this far, then we also test the diagonal wheel cluster pairs to make sure these also conform to the expected shape. In such a case, we have a rectangular shape with each vertex closely matching the expected position of the wheels of a car, so we classify this as a detection $d_{t,i}=\left\{x,y,\phi \right\}$.

These requirements for four clusters all geometrically arranged are reasonably difficult to occur by chance, giving the method a relatively high precision. However, the recall is much lower due to its requirement for four clusters, since the car wheels can be easily occluded by other cars or even by the other wheels of the same car. False LiDAR points, caused by snow and fog, can form false clusters, which decrease the detection precision. This causes the detection performance to deteriorate in poor weather. However, it is a quick and trivial method to implement and allows for us to obtain candidate detections easily. These detections allow us to derive the position of the car as well as the ambiguous orientation (±πrad). Experiments that only required three of the four wheels resulted in a significant increase in misdetections while not contributing much to the overall power of the detection method.

#### 4.3.2. Offline Temporal Filter

To enhance the quality of our annotations, we exploit the temporal domain on top of our clustering-based detection method. Were we to feed the detections alone as annotations, we might expect our neural networks to learn to mimic our clustering-based detector, with all the aforementioned faults. An evaluation of the clustering method shows that while the precision is generally good, the recall statistic is where the method struggles, and this noisy source of training data may impede learning. Therefore, our temporal filter acts to minimise these problems. Note that in this article, we use the terminology of a temporal filter rather than calling it a tracker due to the fact that it is taking into account future information which otherwise would not be available if deployed in a online scenario.

Our offline temporal filter operates on a sliding window principle of sequential frames of length W=50. When there is a detection $d_{t,i}$, we look for any other detections up to t+W in the world coordinate system within a Euclidean distance of ε=0.5 m. If we make this association between a car detected at two different points in time, we can check to make sure that there are also multiple other detections, *R*, within the sliding window. If the car has been detected more than *R* times, then we class it as a high-quality set of detections and apply the temporal interpolation to “fill in” any frames where there was no detection made. In such a scenario, we add an annotation label onto the frame by linearly interpolating between the nearest detections from forwards and backwards in time. Higher-order interpolation methods can be problematic due to the sliding window. The operation of the interpolation can be seen in Algorithm 1.

We can also take advantage of temporal extrapolation, where if we believe a car is present in the data, we can extrapolate the car position beyond the time point of its final detection of the sliding window. If we imagine a stationary car, and the robot drives away from it, we can keep labelling any associated points as belonging to a car well after the clustering method has stopped being able to detect it due to range limitations. Thus, the extrapolation allows our neural networks to receive annotated training samples for vehicles beyond the range of the original method. The advantages here of this process being run offline is that the extrapolation of vehicles can be done into the future and also into the past. The method of operation for the extrapolation is the same as for the interpolation but instead no longer requires both a forwards-in-time and backwards-in-time detection, but only one of those within the window. The requirement of *R* still remains.
**Algorithm 1:** The interpolation operation of our temporal filter, where ε gives our detection distance threshold, *R* is the required number of detections, and W is our sliding window size. Here, lerp is a function that takes two detections at different time points and linearly interpolates the new (x,y,ϕ) at the given time point.
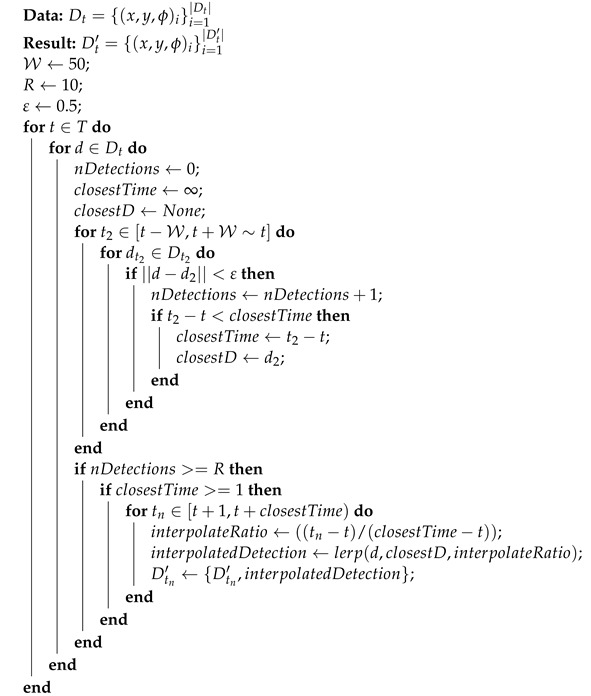


The temporal filter can also be deployed to remove annotation labels in situations where rogue detections are made in isolation. The filter is aware, for example, that for a given detection in a frame, if it has never been seen before and will never be seen again, then it is likely to be a misdetection and therefore can be removed from the training dataset. Non-maximal suppression is also applied spatially.

A precision recall curve was employed to optimise the parameters for the offline temporal filter from a dedicated data collection set. To demonstrate the improvement of the offline temporal filter upon the original detector, we show the performance of a theoretical detector constructed using the temporal filter—note that this is not possible in the real deployment as such a system would have to be able to see into the future. Shown in Figure 5 are the distances from every ground-truth label to the nearest detection and from every detection to the nearest ground-truth label. Table 1 shows the values from this test, demonstrating the benefit of the temporal filter. The results show that for a supposed detection with either method, it is very likely for there to be a ground-truth car nearby, generally within 25 cm of a ground-truth label. Due to the hard constraints of the clustering method, we can see that the clustering method prefers to make safe detections rather than numerous detections, something which is beneficial for training neural networks, as frames without detections can be discarded from the training set.

#### 4.3.3. Weather Simulator Augmentation

In general, the simulation of noise caused by severe weather in the near-infrared ToF LiDAR is a challenging task. Traditional bottom-up approaches based on physical LiDAR models simply do not cover the kaleidoscopic effect caused by various particles in the air. Emerging top-down, data-driven approaches shows extremely promising simulation performance. The weather simulator we opted for as part of our training pipeline was the LaNoising framework (Laser Noising) from [42]. The authors of the simulator tackle three basic problems: (1) data quality, (2) data quantity, and (3) the model training method. The first two problems are strongly connected to the issue of underrepresentation of severe weather conditions in the usual datasets.

The simulator uses a Gaussian process regression model (GPR) to establish whether a particular LiDAR detection would actually be detected in the given weather conditions, and if not, uses a mixture density network (MDN) to apply noise to the detection. Specifically, we consider a set of *N* laser points (i.e., a point cloud) under clear weather conditions, and each point n∈N is processed independently by our noising module. The inputs of this module is a triple $<R,I,V_{vir}>$ formed by the detection range *R*, intensity *I*, and virtual visibility $V_{vir}$, extracted from point *n*, while $V_{vir}$ is the visibility we want to simulate and set manually. The output is the predicted detection range *r*. Since the diffuse reflectors and retro-reflectors do not share the same reflection features, they are distinguished by the intensity *I* (i.e., I<100 for diffuse reflectors, and I≥100 for retro-reflectors according to the Velodyne VLP-32C LiDAR User Manual) and applied with different models. The GPR models predict the distribution of the disappearance visibility in order to decide whether the corresponding ranging process can return the true distance measurements. The MDN models output the detection range distribution if the true object distance cannot be displayed through the first stage.

In such a way, the weather simulator can be directly embedded into the proposed pipeline to apply data augmentation on demand. A visualisation of the weather simulator applying strong fog to the LiDAR scene can be seen in Figure 6. As the training pipeline is run offline, the computational requirements of running the module as part of the training loop are not a significant factor. Nevertheless, the computational speed of the weather simulator amounted to, on average, 14 Hz and therefore increased the *total* dataset processing time by less than 10%.

### 4.4. Neural Network

We chose to apply the Mask R-CNN [46] neural network to the task because the LiDAR data frames are generally visualised as bird’s-eye perspective images (as in Figure 4), and much current state-of-the-art research is focused on image processing neural networks. Mask R-CNN is a well-known and widely used convolution-based object detection neural network. The main feature of the network is that unlike many image-based neural networks that output the bounding box and class labels of detections, it can also output a pixel-wise mask of detected objects, demonstrated in Figure 7. We exploit this particular feature to extract the vehicle orientation in addition to its position.

To train the networks, we used a subset of the data captured in relatively good weather containing approximately 100,000 laser scans collected by the autonomous vehicle described in Section 4.1. These laser scans were processed by the clustering-based detector, and its outputs were passed through the offline temporal filter. The output of the temporal filter was then used to annotate the data, which in turn were then used to train the network. We projected the combined laser scans into a top-down bird’s-eye perspective image and set the pixel values to 1 if its position corresponds to a LiDAR point; otherwise, it was set to 0, thus obtaining a binary image. The annotations also take the form of images, where we take the vector of car detections received from the temporal filter and knowledge of the dimensions of the cars and use this to create a pixel mask of each car instance in the image.

The network was trained for a total of 15 epochs, which was found to be optimal for avoiding over-fitting based on the testing set. The learning rate was set to 0.005 with a momentum of 0.9, step size of 3, and gamma of 0.1. We used a training/testing/evaluation split of 8-1-1, where each unit represents an entire data collection session, to ensure that no temporally close data were at risk of being included in multiple datasets. The training data we used were also augmented with the use of random affine transformations and by applying salt-and-pepper noise with a probability of 0.0001.

Another point about the network training process is that if the pipeline was just left to run, we might expect that much of our training samples would be quite similar. The data collection robot might pause for a few minutes during its operation, for example, accumulating identical training samples at 10 Hz in the meanwhile. Therefore, we imposed a restriction that between each training sample, a *minimum* of 1 m must have been traversed by the robot. This ensures that there are some spatial and temporal changes between every sample. The pipeline is constructed in such a way that this is only applied to the final training samples, and therefore, the temporal filter is still operating at high-frequency in a frame-by-frame manner to ensure maximum information flowing through them about the scene, while only the final training samples bear this spatial constraint.

Once trained, the network provides us with information on the semantic mask of the car detections in the bird’s-eye view. Each mask contains the class score (strength of detection) $s^{i}$, and all pixels with positions p inside the mask are associated with a membership value m(p), indicating how likely they belong to the detected object. To process car detections, we extracted all masks with a class score of $s^{\left(i\right)}>0.9$. For each mask, we consider the centre of its bounding box as the position of the car u,v in the bird’s-eye view image. Since the orientation of the car α is not provided directly from the network, we calculate it separately, taking advantage of the fact that the cars have an elongated shape. First, we extract all pixel positions in the mask with a membership function greater than 0.9 and calculate their covariance matrix, that is, C=cov(p|m(p)>0.9). α is then calculated as $\alpha =atan2(v_{y},v_{x})$, where $v_{y}$ and $v_{x}$ are elements of the dominant eigenvector of C. Finally, we transform the position and orientation of the car u,v,α in the bird’s-eye image into the robot coordinate system, obtaining the position and orientation x,y,ϕ. Naturally, the rotational ambiguity of the detections of π rad remains.

## 5. Experimental Evaluation

In the previous section, we investigated how the robotic hardware acquired and annotated its data and used this to train neural networks. Once training was complete, we experimentally evaluated the trained network. Primarily, we wanted to investigate how beneficial the weather simulations are for improving the robustness of the auto-label-trained neural network. We show the performance of the detector trained both with and without weather simulations and compare them in both good and poor weather. We also perform an evaluation of the rotational accuracy of our models and look at the performance when used with moving vehicles. For evaluation purposes, we set a detection tolerance of 0.5 m relative to the ground-truth labels to classify a correction detection. As it is possible for cars to be heavily occluded but still marked as visible by the annotators, particularly when viewed at oblique angles in the car-park where the experiment was performed, where perhaps only the front bumper is visible, we set a requirement for a minimum number of 15 LiDAR points to be present in a ground-truth label for it to be considered for the false-negative statistic.

As a point of comparison, we also include in our evaluation a method based on the UNet neural network applied to car detection, described in [31], in addition to a comparison to the neural network trained with the same training data hand-annotated. We also adapted the offline temporal filter to work in an online manner to track the cars (i.e., it could no longer take future information into account) to show how the methods perform with real-world tracking applied to the detectors, as is typically the case in real deployments. The tracker is applied exactly as in Section 4.3.2 but no longer able to take anything beyond the current frame into account.

### 5.1. Performance in Good Weather Conditions

To test the performance of the neural networks trained by the proposed method, we used a subset of data captured in relatively good weather containing approximately 100,000 laser scans. In this case, good weather means that there was no fog or precipitation present during data collection. Figure 8 shows the performance achieved by the proposed method in good weather conditions compared to the original clustering detector and Mask R-CNN trained without the aid of weather simulations.

While the auto-label-based approaches perform well for precision, the recall was below 0.8 for all methods, as visible in Table 2. Therefore, we also applied a tracker to the results to give an accurate idea of how the methods would perform in realistic deployment settings.

As the clustering approach requires four known clusters present at specific distances from one another, it is not surprising that its precision is high and also from these strict conditions that the recall is lowest amongst the compared methods. Even with a significant boost from the application of the online tracker, it is still some way behind. The proposed method, *Mask R-CNN + LaNoise*, performs very well in good weather, comparable to the hand-annotated Mask R-CNN and slightly ahead of the auto-labelled Mask R-CNN. However, the cumulative F1-score indicates that the three Mask R-CNN methods all perform very similarly overall, especially when the online tracker has been applied, while the UNet falls a little behind in terms of performance. This gives weight to the idea of auto-labelling providing a viable alternative to the labour-intensive labelling process.

#### 5.1.1. Note on Yaw Estimation

In addition to the detection of the position of the cars, we evaluated the ability of the methods to detect the correct heading of the vehicles. We compared the orientation of correctly detected vehicles with human-annotated ground-truth labels, again for any detections made within 0.5 m of a ground-truth label.

Up to this point, we have considered the orientation of the cars to be ambiguous with respect to the front and rear of the cars, which we have maintained during this analysis. The evaluation showed that the ability to correctly estimate the orientation of detected vehicles was best for the clustering detection method, which with few exceptions was within ±0.1 rad of the ground truth, which can be explained through the fact that it is based on the correctly detected positions of four car wheels. Both the proposed method and the Mask R-CNN trained with human-annotated labels also performed well in this regard, with none but a few anomalous detections across the whole dataset having errors exceeding ±0.2 rad yaw error. In general, all the methods tested were able to reliably determine the orientation of cars within a sufficiently small margin.

#### 5.1.2. Note on Detection of Moving Vehicles

Due to the nature of the LiDAR sensor’s progressive scans, it may not be truly representative to evaluate the performance of the detector on static cars when such robots may realistically encounter moving vehicles during their operations. The progressive nature of the sensor will mean that different parts of the vehicle are captured at slightly different times, and therefore at a slightly different position. This has the potential to distort the vehicle’s shape and therefore impact detector performance. Therefore, to evaluate real-world applicability, we also performed an experiment comparing how the detectors perform in scenarios where both the robot and a car are simultaneously moving around an environment. The results, calculated using approximately 30,000 laser scans, indicated that there was no significant deterioration in performance when detecting moving vehicles.

### 5.2. Performance in Poor Weather Conditions

To test the performance of the proposed method under worsening weather conditions, we collected a dataset during a period of 4 days of heavy snowfall (up to 40 mm/h) and strong wind (up to 12 m/s). This evaluation dataset contains approximately 6000 laser scans; a visualisation of the distortion in a laser scan from these conditions is visible in Figure 4.

Figure 9 shows the performance of the proposed method versus the network trained without the use of the weather simulator-in-the-loop and also against the clustering method from which it was trained. The figure shows that the Mask R-CNN with LaNoise performs as well as clustering in terms of the probability of a ground-truth label being close to supposed detections, while the Mask R-CNN without LaNoise is some way behind. Meanwhile, we can also see that for the probability of a car detection being made close to a ground-truth car, both neural networks perform similarly, some way ahead of the clustering method.

Visible in Table 3, the precision of the methods is much reduced in poor weather. This reflects the higher number of LiDAR points that are now scattered around the scene from ‘mid-air’ detections, which can interfere with the detectors. Meanwhile, generally, the recall statistic is similar to the situation in good weather.

The UNet and auto-labelled Mask R-CNN are particularly affected by this weather, visible in their reduced precision. The hand-annotated Mask R-CNN similarly suffers this drop in precision, while the clustering drop in precision is significant but not as severe. The proposed method performs very well for precision in these conditions. With the application of the tracker, the proposed method is able to maintain a precision of greater than 0.9 in these heavy conditions.

What is significant is how much the neural network-based approaches without LaNoise suffered in poor weather. The much-reduced precision statistic indicates that the weather-induced noise in the LiDAR data was causing them to generate significant false positive detections. It is apparent that they truly must have seen training samples of cars in these conditions to be able to perform well in them.

## 6. Discussion

Our experiments have presented a proof-of-concept auto-labelling pipeline for the detection of cars from a moving car-transporter. In a manner similar to existing auto-labelling methods, we used a relatively straightforward clustering technique to propose candidate detections of cars in our laser scans and made use of a temporal filter to find misdetections and temporally and spatially extend the annotations. These annotations in combination with laser scans were then used as training samples for a neural network for vehicle detection tasks. Our aim was not to design a specialised or task-optimised neural network but to prepare a high-quality training dataset for the existing neural network architecture to learn from.

We define the quality of the training dataset from the perspective of a long-term deployment of an autonomous robot in an outdoor environment. Obviously, the data in any gathered dataset reflects common situations; therefore, the neural networks can learn to create the desired output, and they are known to be very effective. However, during field deployments of a system, problems arise during situations that the system does not expect because they were not observed in advance or were observed rarely. Typical events for an outdoor environment include sudden weather changes and intermittent weather events such as heavy snow, heavy rain, or the sun shining at a low angle—particularly problematic for cameras. Gathering a balanced training dataset that contains a more even representation of different weather events, especially the rare ones, is a time-consuming task that cannot be done in a period shorter than a year. Furthermore, collecting data during inclement weather increases the risk of hardware failure, which increases costs, especially the time needed to gather data. It should be noted that not only dodes hardware resistance limit and slow the process, but the willingness of assistance personnel to work in inclement weather or the ability of people to work in such conditions also prolongs harsh weather data collection. We concluded that collecting balanced data from the perspective of diverse weather conditions requires a longer period of collection than the period needed to observe these rare events.

In contrast to the existing work on the subject of auto-labelling, where the proposed labels are applied directly to an input dataset, we proposed the use of weather simulations as a part of the training pipeline. This enables us to immediately begin generating a relatively balanced training dataset with respect to the expected rare or future weather conditions, something that is often an afterthought. This resulted in a situation where a robot can gather for itself a diverse training set for long-term deployments from a relatively short deployment, freeing up development time for other aspects. In other words, by applying a specialised weather simulator into our pipeline, we removed one of the most labour-intensive parts of training with a bespoke detector for a task while still ensuring the robustness of the final model.

As we were not optimising the neural network architecture *itself* for the task, the performance of the neural network fed with data from the auto-labelling pipeline without weather simulation was slightly worse than the same network fed with manual annotations in good weather. After including the weather simulator, the neural network showed better performance than the classical approach based on human labelling, especially in precision. As the weather conditions deteriorated, the proposed pipeline outperformed the human labelled dataset, showing the strength of the auto-labelling approach, and also outperformed the auto-labelled Mask R-CNN without LaNoise, highlighting the benefit of the application of weather simulations. Importantly, the better performance during harsh weather increases the ability of the robot to perform its services throughout the year with a decreased number of human operator’s interventions and the reduced probability of a critical failure. This increases the ability of the system to gather rare data. We hypothesise that the impact of applying the weather simulator in the pipeline will reduce the overall time needed to gather a good-quality training dataset. This gives us confidence that the simulation-enhanced auto-labelling approach is not only a relatively cheap way of data preprocessing, but it is fundamentally a better and faster approach compared to human labelling.

## 7. Conclusions

Deep neural networks are nowadays a staple in sensor data processing, outperforming hand-coded, engineered methods in terms of precision and recall when tasked with object detection. However, gathering, preprocessing and annotating training datasets for deep neural networks is typically a difficult and labourious task, especially when the training samples are supposed to represent all real-world situations. For outdoor robots, this applies especially to weather conditions, which, in some geographical areas, can vary significantly and change quickly. While the issue of auto-annotating has been addressed by recent works, a representative dataset still requires extensive data collection efforts. To mitigate these issues, we proposed integrating weather simulaton into an auto-annotation pipeline, thereby mitigating the efforts required for both data collection and annotation together. Our pipeline utilises a clustering-based hand-coded method, providing initial candidate detections, which are then enhanced through temporal filtering and tracking. The extra- and inter-polated detections are then passed to the neural network along with the raw data pre-processed by the weather simulation. We have investigated the performance of neural networks, trained by these auto-annotated, simulation-enhanced data, in comparison to neural networks trained on manually annotated datasets.

Our investigation was carried out using an autonomous car transporter operating on a large parking lot over the period of one year, encountering weather conditions from direct sun to rains and blizzards. For the evaluation, we took the most extreme weather conditions encountered and compared the performance of the neural networks trained using our simulation-enhanced auto-annotation pipeline with neural networks trained on hand-annotated data. The results show that the neural networks trained using the proposed method outperformed the ones trained using traditional hand-annotation and auto-labelling without weather simulations. This not only saves significant effort but also opens the path for lifelong learning.

## Figures and Tables

**Figure 1 sensors-22-08855-f001:**
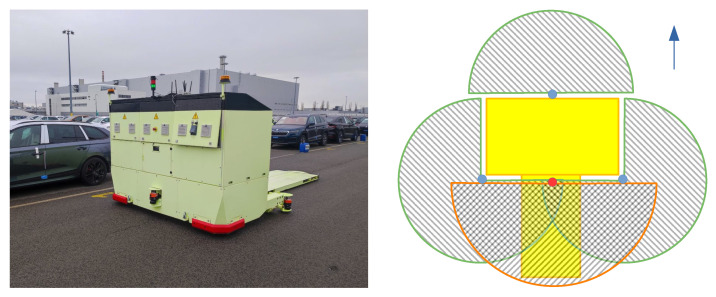
The pipeline was tested on a car transporter visible here in light rain on the left. Also visible are two of the four planar laser scanners positioned at ground level, located at the front, sides, and back of the robot. A visualisation of the positioning of the laser scanners and their field of view relative to the robot can be seen on the right, with the three SICK MICS3 sensors visible in green, and the SICK LMS111 sensor visible in red (sensor range not to scale). The arrow indicates the forward direction of the robot.

**Figure 2 sensors-22-08855-f002:**
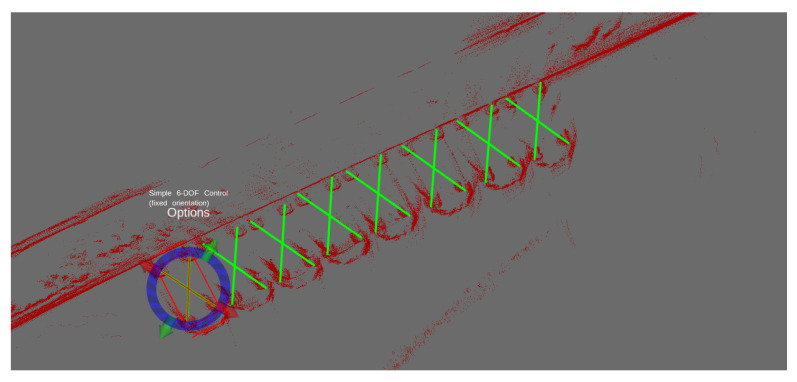
For collecting the ground-truth data, we developed an extension for the ROS RViz program. The laser scans were accumulated on top of each other, and then the real positions of the cars were marked by hand. A second phase, not shown, of annotating the ground truth took the frames individually and allowed the annotators to toggle whether the car was visible at that moment or not.

**Figure 3 sensors-22-08855-f003:**
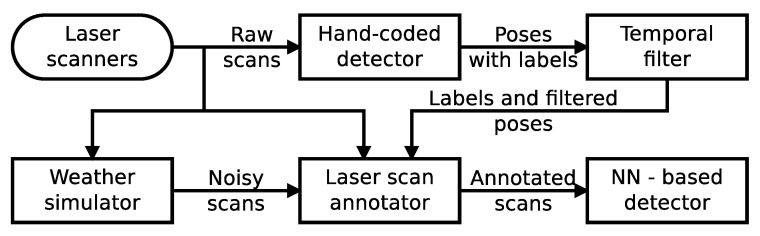
Conceptual model of our auto-labelling pipeline. The laser scans are passed to a clustering-based hand-engineered object detector, whose detections are then corrected through an offline temporal filter. Simultaneously, the laser scans are altered through the weather simulation step. Should the object no longer be detectable in the laser scan due to the weather alterations, or even outright failed to have been detected but was at other time periods, the pipeline can still provide training labels for the current scene to the neural network, allowing it to be trained without human annotation effort.

**Figure 4 sensors-22-08855-f004:**
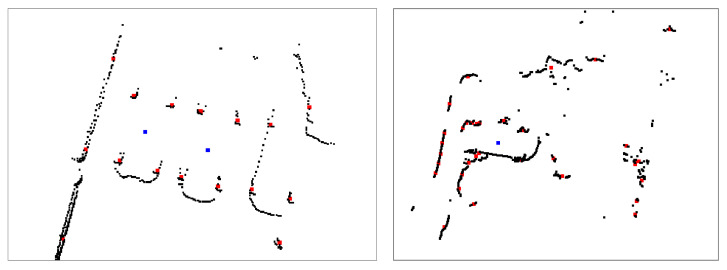
Two images of laser scans captured by the car transporter in different weather conditions. The images are captured by transforming the LiDAR points from each sensor into a common coordinate system, and then projecting the resultant data into a top-down image. In both images, the car transported is positioned off-image, directly below the centre. (**Left**): From clustering and geometry, we can detect cars reasonably well in good weather. The centres of the clusters are shown as red points. The blue points show the centre of the cars as found by the geometrical analysis of the clusters. Note that in ideal conditions, we already have several clusters that appear to not be associated with cars and one car that fails to be detected. (**Right**): Severe weather often affects the reflective properties of light, distorting the data and making the identification of cars difficult, even for human annotators.

**Figure 5 sensors-22-08855-f005:**
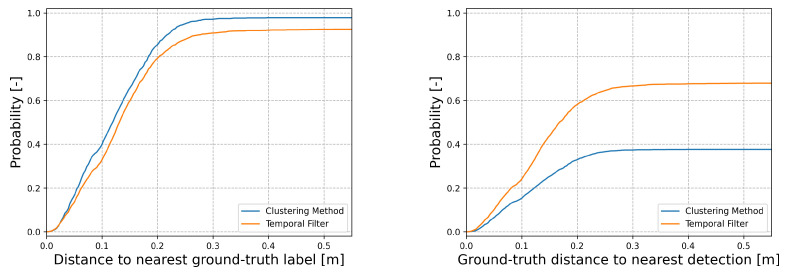
The effect of our offline temporal filter for creating annotations of cars for training networks. In these figures, we show the probability of a ground-truth label being close to a detection (**left**) and the probability of a detection being near a ground-truth label (**right**). Compared to the original clustering-based detector, there is some loss in the precision, which is to be expected from a hand-engineered problem with strict parameters, but the loss of precision is not very high. Unlike the small loss of precision, the temporal filter gains a significant amount of recall compared to the original detector. Note that the temporal filter takes into account future information of detections and therefore only makes sense for offline, supervision-based scenarios.

**Figure 6 sensors-22-08855-f006:**
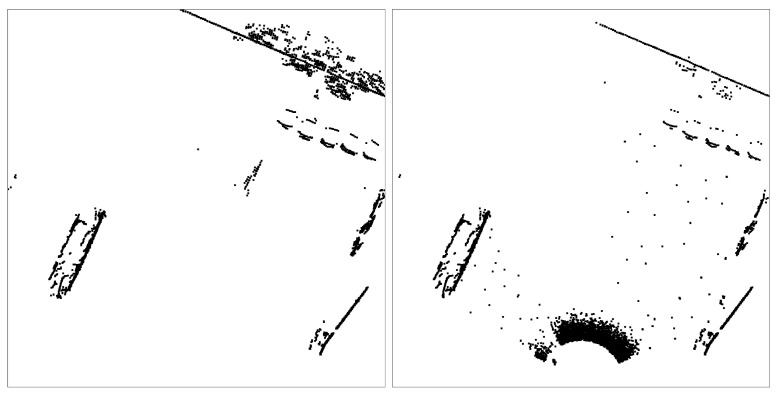
An example of the affects of the weather simulation injecting strong artificial fog on an accumulation of several LiDAR frames, where the laser scanner is positioned at the bottom of the image. (**Left**): the original points captured by the scanner. (**Right**): the affect of the simulations on the data, producing a large band of points close to the sensor, but also produces changes to the appearance of distant objects, reducing the number of usable points for classification. There are also some phantom intermediate detections present. Not visible are the changes to the LiDAR intensity data, which are also modified.

**Figure 7 sensors-22-08855-f007:**
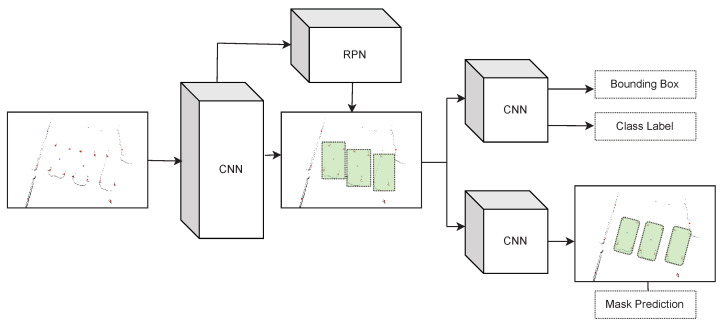
The training pipeline from Figure 3 is used to provide training data and labels to our neural network, in our case Mask R-CNN. The network consists of an initial convolutional neural network (CNN) in combination with a region proposal network (RPN) to identify detection candidates (highlighted in green). The result is run through a further CNN for bounding box prediction and to estimate the class label, and it is run through another CNN for final mask prediction. The masks are projected back to the real-world as defined in Section 4.4. Our pipeline allows for the training process to take place without human annotation effort, and it can be made more robust by our training pipeline’s weather simulations.

**Figure 8 sensors-22-08855-f008:**
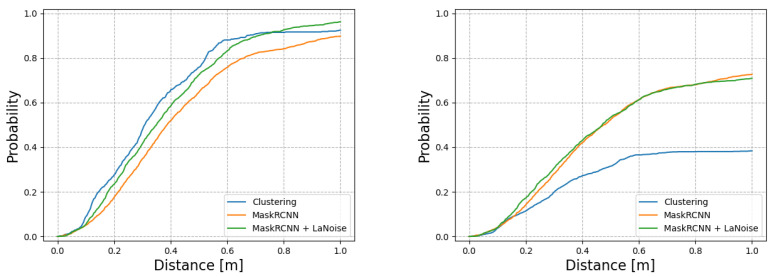
The performance of the detectors in good weather conditions. (**Left**): The probability of there being a car in the ground-truth labels within a certain distance of a detection. (**Right**): The probability of a car detection being made within a certain distance of a ground-truth label. The results show the benefits of using an auto-labelling-based approach to car detection, where approaches both with and without weather simulators outperformed the *clustering method* method from which they were taught.The two neural network-based approaches perform relatively similarly in good weather conditions.

**Figure 9 sensors-22-08855-f009:**
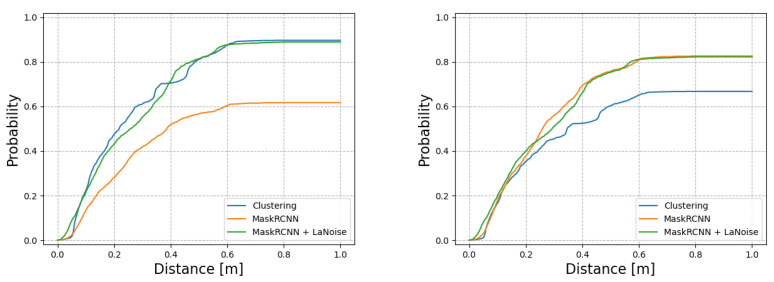
The performance of the detectors in poor weather conditions. (**Left**): The probability of there existing a car in the ground-truth labels within a certain distance of a detection. *Mask R-CNN + LaNoise* and the *clustering method* perform well, while the *Mask R-CNN* on its own suffers during the poor weather. (**Right**): The probability of a car detection being made within a certain distance of a ground-truth label. By this metric, both the neural networks perform well, while the *clustering method* struggles to make detections of cars close to ground-truth labels to the same level.

**Table 1 sensors-22-08855-t001:** The impact of the offline temporal filter on the auto-labelling annotation quality in terms of how many cars were correctly annotated. The filter was assessed on a dedicated evaluation set.

	True Positives	False Positives	False Negatives	Precision	Recall	F1-Score
Clustering Only	292	90	346	0.764	0.457	0.634
Clustering with Temporal Filter	528	129	110	0.803	0.827	0.815

**Table 2 sensors-22-08855-t002:** Comparison of the results of the methods across the good weather dataset. The proposed method, the auto-labelled Mask R-CNN + LaNoise, performed comparatively well to the hand-annotated Mask R-CNN but ahead of the auto-labelled Mask R-CNN.

		Raw			Tracker	
	**Precision**	**Recall**	**F1-Score**	**Precision**	**Recall**	**F1-Score**
Clustering	0.924	0.384	0.542	0.930	0.479	0.632
UNet [31]	0.873	0.645	0.742	0.886	0.696	0.780
Mask R-CNN Hand-Annotated	0.953	0.734	0.829	0.959	0.779	0.859
Mask R-CNN	0.898	0.726	0.803	0.911	0.767	0.833
Mask R-CNN + LaNoise	0.962	0.708	0.816	0.964	0.753	0.846

**Table 3 sensors-22-08855-t003:** The performance of the detectors on data collected during adverse weather conditions. The heavy precipitation causes the recall to suffer across the board, while the proposed Mask R-CNN + LaNoise method maintains a good level of precision, resulting in a significantly improved F1-score compared to the other methods.

		Raw			Tracker	
	**Precision**	**Recall**	**F1-Score**	**Precision**	**Recall**	**F1-Score**
Clustering	0.793	0.568	0.662	0.813	0.604	0.693
UNet [31]	0.672	0.726	0.698	0.700	0.754	0.726
Mask R-CNN Hand-Annotated	0.701	0.751	0.726	0.728	0.777	0.752
Mask R-CNN	0.642	0.748	0.691	0.673	0.774	0.720
Mask R-CNN + LaNoise	0.889	0.744	0.810	0.902	0.772	0.832

## Data Availability

The data and methods presented in this study are publicly available at https://github.com/broughtong/car-detector (accessed on 14 September 2022).

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
