# Peer review of "Embedding Weather Simulation in Auto-Labelling Pipelines Improves Vehicle Detection in Adverse Conditions"

_sensors, 2022, doi:10.3390/s22228855_

Round 1

Reviewer 1 Report

1. The paper title seems to be in disagreement with the content presented in the paper.

2. The schematic block diagram of the proposed methodology needs to be provided to understand the need for deep learning / neural networks.

3. The paper is descriptive to a major extent. It is expected to see conceptual description, mathematical formulation, etc. of the problem in details.

4. Only results are provided. 

4. 

Author Response

Dear reviewer,
We appreciate your interest and taking time to review and provide feedback on the article.
Based on your comments, we have heavily revised the article, and highlighted the associated changes in red for your convienience.
Other colours highlight comments addressing issues raised by other reviewers.
On behalf of myself and the other authors, we thank you for your efforts.

Sincerely, 

George Broughton

=====
Paper title
=====

Reviewer's comment

The paper title seems to be in disagreement with the content presented in the paper.

Authors' answer

Some of the results presented in the paper are not in excess of other methods, that is true.
However, it should be stressed that the method was trained without human labelling, so presenting a method that generally performs on par with much more labour intensive methods may be useful to readers.
I have added a discussion section to the end of the paper in order to try and stress this.
Furthermore, I have extended Table 1 (now table 2) so to present a wider range of results, such as the precision, recall, f1-score, and application of a tracker, indicating that on most metrics the method does indeed perform very well -- even in excess of the teacher in some cases.

=====
Schematic block diagram
=====

Reviewer's comment

The schematic block diagram of the proposed methodology needs to be provided to understand the need for deep learning / neural networks.

Authors' answer

The paper is primarily concerned with the training of the neural networks, covered by the schematic diagram in Figure 2.
To try and stress the need specifically for neural networks I have added the Problem Definition subsection to Section 3, to stress that for the system to learn, we need some general trainable method ie. a neural network.
Additionally, to put the neural network in context, I have added Figure 5 to Section 4 to show how the neural network sits with regard to the rest of the classification task for real-world usage, once it has been trained.

=====
Conceptual description
=====

Reviewer's comment

The paper is descriptive to a major extent. It is expected to see conceptual description, mathematical formulation, etc. of the problem in details.

Authors' answer

Yes this is an excellent point. 
I have heavily rewritten Section 4 ``System Description" in lieu of your comments.
The manipulations of the pointclouds are now explained mathematically, I've added much more detail on the temporal filter, added a pseudo-code algorithm, and added the neural network detection figure (5) in view of this and your last comment.
Finally, I have added in a section titled `Problem Definition'; I hope this fixes your concerns.

=====
Results
=====

Reviewer's comment

Only results are provided.

Authors' answer

I have tried to extent the results section to give more idea to the reader about the impacts of the results.
I have also added a discussion section to the end of the paper to try and give these results in context to existing papers.

=====
Thank you
=====
Thank you again for the time spent with the review. We hope these changes satisfy your concerns. Again, for your convenience, I have highlighted the associated changes for you in red.

Reviewer 2 Report

The study does not seem to be a research paper, there are some observations and results of some methods however, there is no comparison with any other study in the field. You can see some results however there is no clue whether it is good or bad, there is no comparison with any thing

Additionally, neural networks in Section 4.4 is not explained in detail. How the training and test data sets are formed, what the performance is, and what type of models are used, are some questions which have no answer in the study.

Figures 5 and 6 seem to be inconsistent. Fig.5. is “The performance of detectors in regular weather conditions” while Fig.6 is “Comparison of car detection performance during inclement weather”. Firstly, the correlation between precision and recall has not been explained, recall rate is remarkably low for some reason in Fig.6. It is seen to be around 30% which is lower than the one in Fig.5. For such kind of experiments, this does not seem to be a successful value. As mentioned before there is no comparison with any other methodology, the proposal whether a 30% recall value might be low or success is vague.

Author Response

Dear reviewer,
Thank you for your time and feedback on the article.
Based on your comments, we have heavily revised the article, and highlighted the associated changes in green for your convienience.
Other colours highlight comments addressing issues raised by other reviewers.
On behalf of myself and the other authors, we appreciate your efforts.

Sincerely, 

George Broughton

=====
Comparison with other works
=====

Reviewer's comment

You can see some results however there is no clue whether it is good or bad, there is no comparison with any thing
The study does not seem to be a research paper, there are some observations and results of some methods however, there is no comparison with any other study in the field.

Authors' answer

I have added a comparison to a method based on the UNet neural network applied to the task of car recognition.
Unfortunately, most of the similar car detector approaches based on laserscanners mentioned in the related work section are applied to sensors mounted much higher than the one from our data.
Therefore they exploit the ability to detect the "L" shape of a car's contours, which is not present in our data.
Nevertheless, the UNet based approach has been added to the results section to add some comparison.

I have also added tables with the data from the detectors to give a quantitative evaluation of the results, and also added a tracker to try to show real-world performance of the methods.
Additionally the paper has been heavily revised throughout.
I hope these changes address your concerns.

=====
Neural networks details
=====

Reviewer's comment

Additionally, neural networks in Section 4.4 is not explained in detail. 
How the training and test data sets are formed, what the performance is, and what type of models are used, are some questions which have no answer in the study.

Authors' answer

Additional information about how the data is collected for the training process has been added to Section 4.2.
I have changed the text in Section 4.4 and added a more in-depth description about the training hyper-parameters of the network, as well as the training/test set information.

=====
Results
=====

Reviewer's comment
Figures 5 and 6 seem to be inconsistent. Fig.5. is “The performance of detectors in regular weather conditions” while Fig.6 is “Comparison of car detection performance during inclement weather”. Firstly, the correlation between precision and recall has not been explained, recall rate is remarkably low for some reason in Fig.6. It is seen to be around 30% which is lower than the one in Fig.5. For such kind of experiments, this does not seem to be a successful value. As mentioned before there is no comparison with any other methodology, the proposal whether a 30% recall value might be low or success is vague.

Authors' answer
We have re-written the results section to make it clearer.
Regarding the low recall, this was primarily caused by situations where for various reasons a car had relatively few lidar points present in a particular frame (ie. other cars in the car park partially/heavily occluding a car).
In this difficult scenario, the car was generally marked as being visible in the ground truth, despite being only partially visible (as it was a car park, sometimes just the front/back bumper is visible).
Therefore the detector was penalised for missing these in it's recall statistic.

As a solution, we have limited the evaluation to situations where the cars had a minimum number of lidar points visible, and only then performing the evaluation on it.
This means that occluded cars require a certain 'visibility' in order to be counted for the evaluation. 
This is explained in Section 5.

We also showed the performance of the methods when using a online tracker as would be used in real deployments, and added a comparison of the results to a UNet based approach.

=====
Thank you
=====
Again, thank you again for the time spent with the review.
We hope that the revisions satisfy the concerns raised.

Reviewer 3 Report

This manuscript proposes a simulation-enhanced auto-annotation pipeline system, which can automatically collect data to augment the dataset. Meanwhile, the augmented data is pre-processed by the weather simulation, and the labels of the augmented data are automatically annotated instead of hand labeling. The application looks meaningful. However there are some weakness as follows:

1.        The overall system description is not clear. It is mentioned that artificial weather can be added to laser scans. Please provide samples of the augmented scan so that the readers can know the effect of the weather simulator.

2.        System specification is not mentioned clearly based on Fig. 1 and Fig. 2.

3.        Fig. 3 is not explained well. Actually it is not easily to see how come the weather could make influence on laserscars. Meanwhile these two views look like to catch on a top-angle view. It is not clear how the equipment setup as shown in Fig. 1 can make a scan like this.

4.        Auto-label looks an attractive topic and should be an issue in this paper. It is suggested to make clearly discussion and provide some evolution about work on it.

5.        It is suggested to add some diagrams of the architecture of the modules in the proposed system like Mask-RCNN and weather simulator.

6.        In Figures 5 and 6, there is only one method () compared with the proposed algorithm. Is there any statistic of other approaches that could be added to the manuscript?

7.         There are some spelling errors in the manuscript. Please check these mistakes.

Author Response

Dear reviewer,
We appreciate your interest and taking time to review and provide feedback on the article.
Based on your comments, we have heavily revised the article, and hightlighted the associated changes in blue for your convienience.
Other colours highlight comments addressing issues raised by other reviewers.
On behalf of myself and the other authors, we thank you for your efforts.

Sincerely, 

George Broughton

=====
Augmented Scan Visualisation
=====

Reviewer's comment
The overall system description is not clear. It is mentioned that artificial weather can be added to laser scans. Please provide samples of the augmented scan so that the readers can know the effect of the weather simulator.

Authors' answer

This seems like a great idea. We have cleared up the text of the description, and added a visualisation (see Figure 6).

=====
System Description
=====

Reviewer's comment
System specification is not mentioned clearly based on Fig. 1 and Fig. 2.

Authors' answer

We have tried to tidy up the text regarding the system in Section 4.
In regard to this point and from the opinions of the other reviewers, this section has been heavily re-written.

=====
Laserscan to Image Questions
=====

Reviewer's comment
Fig. 3 is not explained well. Actually it is not easily to see how come the weather could make influence on laserscars. Meanwhile these two views look like to catch on a top-angle view. It is not clear how the equipment setup as shown in Fig. 1 can make a scan like this.

Authors' answer

Weather conditions such as fog and snow interfere with the reflectivity of LiDAR beams and distort the reading.
I've added some text to the manuscript mentioning this.
I have also tried to adjust the caption of Figure 3 (now Figure 4) to explain the image better.
You are correct it is a top-down view, where the laserscans have been projected to a common coordinate system and then overlaid onto each other.

=====
Related Work
=====

Reviewer's comment
Auto-label looks an attractive topic and should be an issue in this paper. It is suggested to make clearly discussion and provide some evolution about work on it.

Authors' answer

I have added a few additional references into the related work covering prior work on auto labelling.
These include some older work and some more modern on the subject, concerning using background subtraction in surveillance cameras to extract people, and cross-domain teaching between methods.

=====
Model Diagrams
=====

Reviewer's comment
It is suggested to add some diagrams of the architecture of the modules in the proposed system like Mask-RCNN and weather simulator.

Authors' answer

We have added an image showing the network architecture and processing of an image, see Figure 7.

=====
Method Comparison
=====

Reviewer's comment
In Figures 5 and 6, there is only one method () compared with the proposed algorithm. Is there any statistic of other approaches that could be added to the manuscript?
Authors' answer

We have included a comparison against a similar car detection scheme for planar lidars based on the UNet neural network.
Unfortunately, most of the methods mentioned in the related work are not applied to low-level wheel height LiDARs, but instead to sensors positioned slightly higher, and therefore exploit the fact that the can look for "L" shapes in the data corresponding to the contours of cars.
This means that they wouldn't work on our data.
Therefore we were rather limited so have added a comparison to the UNet approach only.

=====
Spelling
=====

Reviewer's comment

There are some spelling errors in the manuscript. Please check these mistakes.

Authors' answer
We have tried to correct as many as we can find! Thanks for pointing this out.

=====
Thank you
=====

Thank you again for the time spent with the review. We hope these changes satisfy your concerns. Again, for your convenience, I have highlighted the associated changes for you in blue.
